# The Bright and Dark Sides of Resources for Cross-Role Interrupting Behaviors and Work–Family Conflict: Preliminary Multigroup Findings on Remote and Traditional Working

**DOI:** 10.3390/ijerph182212207

**Published:** 2021-11-20

**Authors:** Ferdinando Paolo Santarpia, Laura Borgogni, Chiara Consiglio, Pietro Menatta

**Affiliations:** Department of Psychology, Faculty of Medicine & Psychology Sapienza, University of Rome, via dei Marsi 78, 00185 Rome, Italy; laura.borgogni@uniroma1.it (L.B.); chiara.consiglio@uniroma1.it (C.C.); pietro.menatta@gmail.com (P.M.)

**Keywords:** interruptions, boundary management, resources, remote working, work–family conflict, multigroup, autonomy goal-setting, leadership, proactivity at work

## Abstract

Using boundary management and conservation of resources theories, we examined how job resources (i.e., job autonomy and goal-oriented leadership) and a work-related personal resource (i.e., personal initiative at work) relate to cross-role interrupting behaviors—i.e., interrupting the work (or non-work) role to attend to competing non-work (or work) demands—and how, in turn, they correlate with work–family conflict. Furthermore, we examined differences in the proposed nomological network between workers adopting traditional and remote ways of working. Using a multigroup structural equation modelling approach on a sample of 968 employees from an Italian telecommunications company, we found that: (a) job autonomy was positively related to both work interrupting non-work behaviors and to non-work interrupting work behaviors, (b) goal-oriented leadership was negatively related to non-work interrupting work behaviors, (c) personal initiative at work was positively related to work interrupting non-work behaviors and, finally, (d) cross-role interrupting behaviors were positively related to work–family conflict. Additionally, our findings revealed previously undocumented results; (a) mediating patterns in how resources relate, through cross-role interrupting behaviors, to work–family conflict and (b) non-invariant associations among job autonomy, cross-role interrupting behaviors and work–family conflict across traditional and remote workers. The limitations and theoretical and practical implications of the present study are discussed.

## 1. Introduction

Information and communication technologies (ICTs) have enabled organizations to adopt remote and flexible working arrangements (referred to as “new ways of working”), providing employees greater opportunities to decide where, when and how to work [1]. However, ICTs and new ways of working have already shown contradictory effects on employees’ overall well-being [2], both granting benefits, such as shorter commuting time and schedule flexibility, but also posing new risks due to work intensification and more frequent interferences between work and one’s personal life [3,4]. As the COVID-19 outbreak dramatically accelerated digital transformation processes in organizations [5], calls for special attention to its effects in the construction of a theoretical framework that allows acting upon the “new normal” is needed.

One perspective in looking at these ambiguous effects is provided by the increasing blurring of work–family boundaries, making each domain more pliable and permeable to experiences concerning the other [6]. Indeed, the looseness of boundaries eases the occurrence of cross-role interrupting behaviors, that is, interrupting a work (or non-work) role to attend to competing non-work (or work) demands [7]. As a result of being “always on”, behaviors such as checking work e-mails during leisure time or, vice versa, texting with a family member while working can be quite common during a regular day [8]. Cross-role interrupting behaviors, however, exert disruptive effects on individual role-specific performance and well-being due to cognitive (e.g., attentional conflict), affective (e.g., negative emotional responses) and self-regulatory (e.g., goal obstruction) costs [9]. As such, the literature has already highlighted that cross-role interrupting behaviors act as powerful triggers to work–family conflict [10] that is, an experience of inter-role interference and incompatibility between domains that hinders role performance and well-being [11].

Accordingly, Boundary Management theories have become central to understanding how individuals deal with interruptions arising from work and non-work domains to achieve or maintain their balance [12]. Although boundary management is determined by both personal and contextual characteristics [7], growing literature on the antecedents of cross-role interrupting behaviors has not yet deepened the role of resources in the facilitation or inhibition of their occurrence. Resources refer to personal and contextual assets that can foster individual functioning and buffer the detrimental effects of demanding contextual features [13]. As such, resources are crucial to enable work-life balance [14], as they facilitate work–family relationships [15], while reducing inter-role conflict [16]. In contrast, few studies have found that some resources may have the downside of increasing role blurring that, in turn, drains the beneficial effects on work–family conflict [17,18]. Integrating work–family boundary management [7,19,20] and conservation of resource [13] perspectives, we argue that cross-role interrupting behaviors, as a boundary blending and resource depleting mechanism [9], may address this potential downside of resources in the relationship with work–family conflict. Specifically, we chose to investigate the enabling and depleting pathways to work–family conflict, through interrupting behaviors, of job autonomy, goal-oriented leadership style and personal initiative at work. Indeed, job autonomy, as a job resource that enables more control over one’s work, has already proven to be fundamental for reducing work–family conflict [16].

However, it has recently been associated with work-life intensification [21] and role blurring [18]. Instead, the literature abounds with contributions on the role of positive leadership styles as a provider for a set of psychological support to employees in ameliorating work–family conflict experiences [16]. However, little is known on the role of leadership in work–family outcomes by shaping employees’ boundary management behaviors [20]. In the present study, we argue that goal-oriented leadership, as an empowering and supportive style to goal achievement [22,23], may help employees filter out interruptions arising from the family context. Furthermore, although typically associated with positive outcomes [24], research has found that personal initiative at work, as a set of proactive behaviors that exceed work role requirements [25], may deplete resources investable in the family domain to the point of generating work-to-family conflict [26]. Finally, considering that previous meta-analytical results have provided mixed support for the potential negative effects of boundary-blurring on flexible and remote workers [27,28], mechanisms explaining the pros and cons of new ways of working on work–life balance are still not adequately understood.

The present research aims to contribute to the literature in several ways. First, we investigated the specific relationships between two job resources (i.e., job autonomy and goal-oriented leadership), one work-related personal resource (i.e., personal initiative at work) and cross-role interrupting behaviors, considered as a bidirectional construct (i.e., work interrupting non-work behaviors and non-work interrupting work behaviors) [7]. Second, we extended the analysis of the role of the two different directions of cross-role interrupting behaviors on the occurrence of work–family conflict, also investigated as a bidirectional construct (i.e., work-to-family conflict and family-to-work conflict) [14]. Third, the study deepens the bright and dark sides of resources in work–family boundary management through the mediating role of cross-role interrupting behaviors in the relationships between resources and work–family conflict. Finally, differences occurring across groups of workers differing in their work arrangement (remote working vs. traditional working) within the proposed conceptual model are tested. Therefore, a cross-sectional and multigroup design was proposed to verify the hypothesized relationships. Figure 1 represents the overarching model that guides the present study.

### 1.1. The Conceptual Framework: Boundary Management and COR Theories

Individuals are embedded in complex social contexts and, throughout their lives, take on multiple social roles. Among these, work and family domains significantly shape the construction of individual identity, so that being able to successfully meet role expectations arising from both domains strongly contributes to personal fulfillment [7,19,20]. Through work and family roles, individuals allow work and family contexts to reciprocally interact as open social systems (i.e., work–family interface), making the occurrence of material, energetic and informative exchanges possible (i.e., spillover) [29]. However, in order to exist as clearly differentiated entities, work and family contexts and roles must have some limits, namely boundaries, defining the degree and the quality of their interconnection [7,19,20]. Boundaries can be both physical (i.e., defining the degree of flexibility about where and when work and family activities are done) and psychological (i.e., defining the degree of permeability across roles’ affective, cognitive and behavioral patterns) [20]. Moreover, boundaries are socially constructed, which means that individuals actively negotiate with work and family domain boundary-keepers (e.g., supervisor at work or spouse/partner within the family) in order to identify where boundaries lie and the ease with which individuals can cross them to deal with competing role demands [12], in a way that is meaningful to their identity [7].

Although cross-role transitions occur on a daily basis and are typically routinized over time, unexpected events may force such boundary blending experiences [19]. Indeed, individuals often find themselves juggling between multiple demands arising from both domains and interrupting one role to attend to another role, namely cross-role interrupting behaviors [7]. Cross-role interrupting behaviors can be triggered by external cues (e.g., intrusive, demanding or distracting social signals initiated by another context), as well as internal cues (e.g., physiological needs, mood, fatigue, concerns associated with another role) [30]. As such, the literature differentiates Work interrupting Non-Work behaviors (W→NW; e.g., checking work e-mails while engaged in a domestic task) from Non-Work interrupting Work behaviors (NW→W; e.g., answering a phone call from one’s spouse on the job; [7]).

Cross-role interrupting behaviors are typically considered to be inducive of strain reactions [1] as they involve several cognitive, affective and self-regulatory costs. Specifically, once engaged in cross-role interrupting behaviors, employees are challenged with a primary cognitive and behavioral effort intensification to meet the interrupting demand and negative affective reactions, but also with an additional effort to resume or reschedule the interrupted task (i.e., resumption lag), which in turn, negatively affects role performance and well-being [9]. Indeed, according to COR theory [13], since individuals are intrinsically motivated to preserve or expand current resources, the strain will emerge whenever they are lost and threatened or when resource investments exceed their returns. Resources, when applied to work–family relationships, are personal or contextual (either work or family-related) valuable assets that improve work and home functioning by facilitating gain processes, in which contextual and personal resources accumulate (i.e., work–family enrichment), and buffer loss processes, in which stress develops due to contextual demands and personal resource depletion (i.e., work–family conflict) [14]. 

As such, job and personal features, valuable as resources, should limit or prevent interrupting behaviors. Instead, drawing on Schieman and Young’s role blurring hypothesis [18], resources that facilitate boundary blending experiences have a potential downside: although easing the transition between roles can reduce the risk of work–family conflict, it also increases exposure to interruptions which, instead, are likely to activate a depleting pathway towards work–family strain. Therefore, we argue that cross-role interrupting behaviors may be a mechanism that explains the downside of resources. Below, this potential double-faced path of resources towards work–family conflict, through cross-role interrupting behaviors, is specifically explored for job autonomy, goal-oriented leadership and personal initiative at work.

#### 1.1.1. Cross-Role Interrupting Behaviors and Work–Family Conflict

Work–family conflict is a form of inter-role conflict in which work and family domains are mutually incompatible or interferent with each other [11]. Such inter-role conflict is primarily caused by the fact that the demands associated with one role may drain the time at someone’s disposable (i.e., time-based conflict) and/or solicit a psychological tension (i.e., strain-based conflict), such that the fulfilment of the demands associated with another role is hindered [11]. Secondly, work and family roles can require a set of competing behaviors which are distinctly appropriate within domains rather than across them (i.e., behavior-based conflict) [11]. Moreover, similar to cross-role interrupting behaviors, work–family conflict is also characterized by bidirectionality [14]. Specifically, Work-to-Family conflict (W→F) is a process whereby work demands (e.g., meeting the project deadline) interfere with one’s functioning in the family context (e.g., marital satisfaction); while Family-to-Work conflict (F→W) is a process whereby family demands (e.g., providing care to dependent children) interfere with one’s functioning in the work context (e.g., performance) [14].

Due to their costs, it can be argued that cross-role interrupting behaviors may trigger work–family conflict. Indeed, individuals deal with demanding cognitive, affective and self-regulatory processes, such as enacting the appropriate role behavioral script for the interrupting domain [19] and engaging in time and energy-consuming activities for the interrupted domain [9]. Several studies have already supported this assumption [10,31,32,33]. Specifically, employees that frequently use communication technologies (e.g., e-mail; cell phones) to perform their job during non-work hours also report higher W→F conflict [31,33], partly due to negative emotional reactions (i.e., anger) [32]. Furthermore, the more the occurrence of boundary violations at home (or work) is perceived as a family (or work) goal-related obstruction, the more individuals have negative emotional reactions and experience work–family conflict in a directionally congruent fashion [10]. As such, we formulated the following hypotheses:

**Hypothesis** **1** **(H1).**
*NW→W interrupting behaviors are positively related to F→W Conflict.*


**Hypothesis** **2** **(H2).**
*W→NW interrupting behaviors are positively related to W→F Conflict.*


#### 1.1.2. The Relationships between Job Autonomy, Cross-Role Interrupting Behaviors and Work–Family Conflict

Autonomy reflects the extent to which a job allows freedom, independence and discretion to decide and choose how, when and where to carry out work tasks [34,35]. Although traditionally considered as a job resource that positively influences work-related outcomes such as well-being and performance, research has also highlighted its drawbacks [21]. Indeed, under such empowering circumstances, employees are challenged with increased accountability, need to control and schedule work which can lead them to intensify their efforts [36]. If this is associated with an intensification of work (i.e., facing a faster work pace, with decreased leisure time and increased multitasking) [37], job autonomy may favor strain experiences [21]. In a work–family boundary management perspective, it might be expected that job autonomy allows more control over one’s work-non-work boundaries and the flexibly enactment of cross-domain transitions to attend multiple role expectations [17]. However, Schieman and Young [18] found that schedule control (i.e., a specific component of job autonomy concerning the degree of control over work temporal parameters) significantly increases role blurring in the form of work–family multitasking. Moreover, such role blurring may be often required by externally set and unexpected demands interfering with personal schedules [21] of which domain-specific members (e.g., colleagues; relatives) may not be sufficiently aware of [20]. Therefore, it can be argued that, since job autonomy gives the opportunity of flexibly scheduling work around family activities and vice versa [17,18], it may loosen boundaries to match competing role demands as they occur during hours traditionally reserved for work or family time. Based on this, we hypothesize that:

**Hypothesis** **3** **(H3).**
*Job autonomy is positively related to (a) NW→W interrupting behaviors and (b) W→NW interrupting behaviors.*


Past research has shown that job autonomy is an effective resource in reducing work–family conflict [16]. However, job autonomy can also drain employee’s energy and time through inter-role blurring, which instead may easily lead to work and family domain conflicts [18]. Schieman and Young [18] found that the positive association between schedule control and work–family multitasking suppresses the negative association between schedule control and work–family conflict. Therefore, it could be argued that if job autonomy eases the enactment of cross-role interrupting behaviors, then such a resource depleting experience may lead to inter-role conflict. Hence, the following hypothesis was investigated:

**Hypothesis** **4** **(H4).**
*Cross-role interrupting behaviors mediate the relationship between job autonomy and work–family conflict.*


#### 1.1.3. The Relationships between Goal-Oriented Leadership, Cross-Role Interrupting Behaviors and Work–family Conflict

Goal-oriented leadership, derived from goal-setting theory [23] can be defined as a positive leadership style characterized by the assignment of clear, specific, and challenging goals to the employee, trusting his/her autonomous ability to reach them and providing constructive support and feedback during the process [22]. Within the work–family literature, goal-oriented leadership may be conceived as a form of general supervisor support, which is intended to enhance personal effectiveness at work and must be distinguished from supervisor’s work–family supportive behaviors, which specifically facilitate the employee inter-role balance [38]. Since work–family boundaries are negotiated between the subordinate and his/her supervisor, as a crucial boundary-keeper for the work domain [20], employees’ perception of what the supervisor pays most attention to may model their boundary management behaviors [39]. Specifically, the ease of domain-crossing for employees largely depends on the supervisor’s awareness about their family needs and duties and availability to craftwork around them [38]. Indeed, Ferguson, Carlson and Kacmar [40] found that the supervisor’s family-supportive behaviors are associated with the perceived ability of the employee to stretch work boundaries to meet family demands as they arise during worktime. Instead, the perception of a goal-oriented leader who provides resources oriented towards goal accomplishment, may help the employee to guard work boundaries from interrupting family demands, thereby making cross-role interruptions less behaviorally available [19]. Consequently, we formulated the following hypothesis:

**Hypothesis** **5** **(H5).**
*Goal-oriented leadership is negatively related to NW→W interrupting behaviors.*


When employees feel supported at work by their supervisor, they gain additional job resources (e.g., skills, positive emotions, different perspectives) that facilitate the reconciliation between work and family domains, both enriching their relationships [15] and buffering the strain caused by inter-role conflict [16]. Since NW→W interrupting behaviors are likely to deplete resources required to participate in the work role (Matthews et al., 2014), employees with greater access to a goal-oriented leader may gain additional personal psychological resources (e.g., decreased stressor appraisal; increased concentration) [14] to effectively filter out non-work interruptions on work goals and, therefore, prevent inter-role conflict initiated from the family domain. Hence, the following hypothesis was investigated:

**Hypothesis** **6** **(H6).**
*NW→W interrupting behaviors mediate the relationships between goal-oriented leadership and F→W conflict.*


#### 1.1.4. The Relationships between Personal Initiative at Work, Cross-Role Interrupting Behaviors and Work–Family Conflict

Personal initiative is a “behavior syndrome resulting in an individual’s taking on an active and self-starting approach to work and going beyond what is formally required in a given job” [25] (p. 140). In other terms, the personal initiative reflects an individual propensity to engage in a set of proactive actions at work that exceeds work role requirements (e.g., anticipating future work demands; actively and persistently overcoming setbacks to work goals) [25]. Prior research on this personal resource has already highlighted its positive influences on individual and organizational outcomes, such as performance [41]. Indeed, proactivity enables motivational pathways to individuals’ optimal and healthy functioning [24] and serves the purpose of expressing and nourishing one’s self-identity [42]. However, scholars have recently pointed out the dark side of proactive behaviors at work as they can often require extra-role efforts, such as working extra hours or taking on additional responsibilities, which are likely to deplete resources otherwise investable in family or personal activities [43]. When individuals are managing different expectations arising from multiple roles, they choose to engage in role-taking behaviors that are salient to their identities [7]. Since individuals are intrinsically motivated to expand what they value the most [13] and since the work domain may be particularly central in modeling employee’s identity when they are characterized by higher personal initiative [42], it could be argued that this should reflect upon boundary dynamics. That is, it may make work role-related stimuli motivationally appealing and the object of higher effort investment [24], leaving family boundaries more pliable to violations and trespassing (e.g., integrating work activities during leisure time) [19,20]. This assumption consequently implies that personal initiative at work may increase chances of letting the work role easily or frequently interrupt the family role (rather than vice versa). As such, we formulated the following hypothesis:

**Hypothesis** **7** **(H7).**
*Personal initiative at work is positively related to W→NW interrupting behaviors.*


Proactivity at work increases employee role demands so that additional resource investment is required [25] and a straining process may be activated [24,43]. Indeed, research has found that proactivity at work is positively related to daily salivary cortisol and bedtime fatigue [44], affective strain [45], rumination and interpersonal withdrawal [46]. The resource depletion pathway of proactivity also seems to impact the reconciliation of work and family roles. Specifically, Bolino and Turnley [25] found that higher levels of the individual initiative at work are associated with higher levels of role overload, job stress and W→F conflict. Therefore, if proactivity at work makes employees more prone to let the work role interrupt the family role, then such behavior may, in turn, deplete resources required to participate in the family domain, thereby increasing chances of W→F conflict occurring. As such, the following hypothesis was investigated:

**Hypothesis** **8** **(H8).**
*W→NW interrupting behaviors mediate the relationship between personal initiative and W→F conflict.*


#### 1.1.5. Differences across Remote and Traditional Workers: The Moderating Role of Working Arrangement

According to the Italian normative context, remote and flexible working (labeled as smart working) is defined as a way of executing an employment relationship established by agreement between parties, with forms of organization by phases, cycles and goals and without precise time or spatial constraints, with the possible use of technological tools for carrying out the work activity (Law Decree DL-81/2017–Check for more information: https://www.normattiva.it/do/atto/export (accessed on 17 June 2021)). Although there are widely diverse normative definitions across countries [28], remote and flexible working falls into the new ways of working (NWW) spectrum that spans work designs varying in the degree to which allow employees to control “the timing and place of their work, while being supported by electronic communication” [1] (p.114). Research has found opposing yet co-existing effects of NWW on remote workers’ work-life balance [2]. Indeed, although enabling multiple role expectations in personally suitable ways [27], NWW may cause strain by exposing employees to interruptions and social overload [8]. Since working conditions may structurally influence role transitions [7], we argue that remote working may enhance the chance to engage in cross-role interrupting behaviors and, therefore, exacerbate their relationship with work–family conflict. Furthermore, NWW affects employees by shaping their perceptions of work context features [6], so it is likely that they can interact with each of the resources considered in the present study. First, job autonomy can counterintuitively foster work intensification and/or organizational control over one’s life is associated with informal practices that endorse increased work effort (i.e., autonomy paradox) [47]. These paradoxical effects should be enhanced in work settings that are highly flexible and characterized by indirect control rather than traditional work contexts [21]. Therefore, if remote workers maximize their autonomy over work schedules and locations [27], then they may experience interruptions more intensely [1] that, in turn, easily put domains in conflict [18]. Second, leadership behaviors based on the principles of autonomy, trust and responsibility are regarded as fundamental to the implementation of NWW [48]. Moreover, research suggests that, on days of remote working, employees may have more effective communications for goal accomplishment and quicker access to feedback [8], which favor their engagement while reducing strain during work tasks [1]. Therefore, if employees can simultaneously benefit from a goal-oriented leader [22] and remote and flexible work arrangements, the hypothesized positive effects of such leadership style in preventing cross-role interrupting behaviors and, in turn, work–family strain may be boosted. Third, research has suggested that proactive employees are likely to devote fewer resources to family obligations by going beyond the call of duty for their employer [43]. Since employees are prone to reciprocate the flexibility that is offered to them by intensifying their efforts [36], we argue that remote workers with higher personal initiative may more easily stretch non-work boundaries to this end and, therefore, experience an enhanced inter-role conflict. All in all, since remote workers are socially embedded in a more boundaryless context than traditional workers, then it could be argued that the proposed model may vary across groups of workers differing in their working arrangement. Based on this, the following research objective was investigated:

**Hypothesis** **9** **(H9).**
*Working arrangement (remote vs. traditional working) moderates the hypothesized nomological network.*


## 2. Materials and Methods

The study was conducted via an online self-report questionnaire before the COVID-19 emergency in a large Italian telecommunications company. Participation in the study was voluntary, and the research team guaranteed anonymity to all respondents. The final sample comprised 968 employees, 66.33% remote workers and 33.67% traditional workers. Seventy-five percent of the sample were male. Regarding age, 65% were born between 1965 and 1980 (ranging from 40 years old to 55 years old), 31% were born before 1965, and 4% after 1980. Regarding educational level, 30% of participants had a bachelor’s or master’s degree, while 70% qualified lower than a bachelor’s degree. In relation to a professional qualification, 88% of the participants were white-collar workers, while 12% were middle managers. The subsamples of workers (remote vs. traditional ones) were homogeneous in terms of socio-demographic and organizational variables.

### 2.1. Measures

#### 2.1.1. Job Autonomy

Job autonomy was assessed by using six items adapted from Morgeson and Humphrey [34]. The items specifically measure the degree of worker’s discretion in scheduling work and choosing when, where and how to do the job (e.g., *“In my company I can decide where to work”*; *“I can decide how much time to allocate to a specific activity”*). Participants rated their agreement on a 6-point Likert scale ranging from 1 (*“strongly disagree”*) to 6 (*“strongly agree”*). The coefficient alpha reliability was 0.88.

#### 2.1.2. Goal-Oriented Leadership

Goal-oriented leadership was assessed by using six items adapted from Borgogni and Dello Russo [22] and drawn from Locke and Latham [23]. Items specifically measure the degree to which the supervisor is perceived as supportive to goal achievement through behaviors such as crafting goals over employees’ characteristics, empowering them over the choice of strategies and providing support in the face of obstacles (e.g., *“My supervisor assigns specific and targeted goals for me”*; *“My supervisor updates me regularly on the progress towards the goal”*). Participants rated their agreement on a six-point Likert scale ranging from 1 (*“strongly disagree”*) to 6 (*“strongly agree”*). The coefficient alpha reliability was 0.85.

#### 2.1.3. Personal Initiative at Work

Personal initiative at work, assessing employee’s proactive approach to work and beyond what is formally required was measured through a seven-item scale from Frese and colleagues [25] (e.g., *“I take initiative immediately even when others don’t”* and *“Usually I do more than I am asked to do”*). Participants rated their agreement on a 5-point Likert scale, ranging from 1 (*“never”*) to 5 (*“always”*). The coefficient alpha reliability was 0.80.

#### 2.1.4. Cross-Role Interrupting Behaviors

Cross-role interrupting behaviors, describing the degree to which individuals engage in cross-domain boundary crossing interrupting behaviors, was assessed through an eight-item scale adapted from Kossek and colleagues [7] and measuring two factors: W→NW interrupting behaviors (5 items) and NW→W interrupting behaviors (3 items). An example of a W→NW interrupting behaviors item is: *“I respond to work-related communications (e.g., emails, texts, and phone calls) during my personal time away from work”*; while an example of a NW→W interrupting behaviors item is: *“I take care of personal or family needs during work”*. Participants rated their agreement on a six-point Likert scale, ranging from 1 (*“never*”) to 6 (*“always*”). The coefficient alpha reliability was 0.85 for W→NW interrupting behaviors and 0.72 for NW→W interrupting behaviors.

#### 2.1.5. Work–Family Conflict

Work–family conflict, describing the degree to which individuals perceive interference and incompatibility between work and family domains, was assessed through a nine-item scale adapted from Netemeyer and colleagues [49] and measuring two factors: W→F conflict (4 items) and F→W conflict (5 items). An example of a W→F conflict item is: *“The amount of time my job takes up makes it difficult to fulfil my family responsibilities**”*; while an example of a F→W conflict item is: *“The demands of my family interfere with work-related activities”*. Participants rated their agreement on a six-point Likert scale, ranging from 1 (*“never”*) to 6 (*“always”*). The coefficient alpha reliability was 0.85 for W→F conflict and 0.74 for F→W conflict.

#### 2.1.6. Working Arrangement

Working arrangement was assessed using a single question *(“Are you currently making use of smart working?”*) measuring a dichotomous variable (remote working was coded as 1, traditional working as 2).

### 2.2. Statistical Analysis

Prior to hypothesis testing, the distinctiveness of the hypothesized latent variables (i.e., job autonomy, goal-oriented leadership, personal initiative at work, W→NW interrupting behaviors, NW→W interrupting behaviors, W→F conflict and F→W conflict) were investigated by running a confirmatory factor analysis (CFA), using the maximum likelihood (ML) estimator in Mplus 8.1 [50]. The appropriateness of the model fit was established with (1) values of CFI higher than 0.90 [51] and (2) RMSEA values of 0.08 or less with associated confidence intervals [52]. Then, to properly determine whether the hypothesized seven-factor model showed the best fit to the data, it was compared with plausible competitive models differing in their factorial structure. Specifically, a seven-factor model was tested against:A first alternative five-factor model (M1a), in which the latent factors of W→NW and NW→W interrupting behaviors were merged, as well as W→F and F→W conflict was tested. The two resulting factors were correlated with three distinct factors representing job autonomy, goal-oriented leadership and personal initiative at work. This model investigates whether two latent factors are enough to explain scores on W→NW and NW→W interrupting behaviors, on the one hand, and on W→F and F→W conflict, on the other.A competing alternative five-factor model (M1b), in which the latent factors of W→NW interrupting behaviors and W→F conflict were merged, while NW→W interrupting behaviors were merged with F→W conflict was tested. Basically, this model investigated the distinctiveness of directionally congruent work-non-work latent dimensions.The last alternative model (M2) implements Harman’s single factor test [53], using a model in which all items from all constructs in the study load on a single factor to exclude the influences of method bias on observed item covariances [54].

The best measurement model was evaluated, for each comparison, through fit indices and significant differences in chi-square values (Δχ^2^; *p* < 0.001) [55]. Following this, the main hypotheses were tested with multi-group confirmative factor analysis (MGCFA) within the structural equation modeling (SEM) framework.

According to suggested procedures [55], we first tested measurement invariance and once supported, structural invariance. In regards to measurement invariance, we tested the goodness of fit of separate baseline models for each group. Then, the model invariance was tested across groups at three increasingly stringent levels [56]: (1) configural invariance (i.e., no equality constraints are imposed on parameters); (2) metric invariance (i.e., factor loading equality constraints are specified across groups); and (3) scalar invariance (i.e., item intercept equality constraints are specified across groups). At each step, the model’s goodness of fit was evaluated and the significance of measurement invariance was assessed through model differences in comparative fit index (ΔCFI) with values lower than −0.01, paired with changes in RMSEA of 0.015 and SRMR of 0.030 (for metric invariance) or 0.015 (for scalar invariance) [56]. With respect to structural invariance, again we tested the goodness of fit of separate structural baseline models for each group. Specifically, we tested a full-mediation model in which all possible coefficient paths were estimated and cross-role interrupting behaviors mediated the relationships between resources and work–family conflict, using the robust maximum likelihood estimator (MLR, which is robust to non-normality and non-independence of observations but produces equivalent estimates as standard ML) [57].

In line with previous studies [15,16], gender, age, education level and professional qualification were employed as controls. Then, multigroup structural models were tested, initially without any constraints on coefficient paths across groups (i.e., unconstrained model) and then, imposing equality constraints across groups (i.e., constrained model). Again, the goodness of fit of each model was evaluated. To assess the significance of structural invariance, models were compared by using the DIFFTEST option in Mplus–Check for more information: https://www.statmodel.com/chidiff.shtml (accessed on 1 September 2021), which allows models’ comparison when MLR is used through the employment of the Satorra and Bentler [58] scaled chi-square statistic and standard errors (SB χ^2^) and ‘the scaled difference chi-square’ test (ΔSBχ^2^). 

When the difference between models’ chi-square statistics is significant, this means that some paths are different across the groups analyzed [55]. Following the procedure employed by several other previous studies [59,60,61,62], we performed an iterative process to assess invariance for each of the structural coefficient paths separately to inspect the location of non-invariance. Specifically, iterative tests were performed by progressively adding one constrain in a specific path. If the comparison between the unconstrained model and the new model is not significant, then non-invariance across groups in this specific path is supported. This process was repeated until we reached the final model. The significance of the indirect effects of the final model were investigated by estimating their standardized estimates and associated 95% confidence intervals in using the RMediation package in R 3.6.0 (R Development Core Team, Wien, Austria) [63,64].

## 3. Results

Descriptive statistics and zero-order correlations among job autonomy, goal-oriented leadership, personal initiative at work, W→NW interrupting behaviors and NW→W interrupting behaviors, W→F conflict and F→W conflict, gender, age, education and professional qualification are listed in Table 1. As expected, resources showed a differential correlational pattern with cross-role interrupting behaviors. Specifically, job autonomy was positively and significantly correlated with both W→NW interrupting behaviors (ρ = 0.193) and NW→W interrupting behaviors (ρ = 0.148). Contrary to our expectations, Goal-oriented leadership positively and significantly correlated with W→NW interrupting behaviors (ρ = 0.142). Personal initiative at work positively and significantly correlated with NW→W interrupting behaviors (ρ = 0.331). In turn, NW→W interrupting behaviors were positively and significantly correlated with F→W conflict (ρ = 0.349) and W→F conflict (ρ = 0.178). W→NW interrupting behaviors were positively and significantly correlated with W→F conflict (ρ = 0.510) and F→W conflict (ρ = 0.146). Correlations were stronger for directionally congruent measures.

### 3.1. Confirmatory Factor Analysis

Empirical tests (see Table 2) supported the hypothesized seven-factor model (M0), which indeed showed good fit indices and clearly fit the data better than any alternative model considered. Specifically, the comparison between the hypothesized seven-factor model and the first alternative five-factor model (M1a) supported the factorial distinctiveness of W→NW and NW→W interrupting behaviors, on the one hand, and of W→F and F→W conflict, on the other (Δχ^2^ (11) = 1207.693, *p* < 0.001). Moreover, the comparison between the hypothesized seven-factor model and the second alternative five-factor model (M1b) supported the factorial distinctiveness of W→NW interrupting behaviors and W→F conflict, on the one hand, and of NW→W interrupting behaviors and F→W conflict, on the other (Δχ^2^ (11) = 1404.249, *p* < 0.001). Finally, results clearly rejected the single factor model (M2; Δχ^2^ (21) = 8892.301, *p* < 0.001). The factor loadings were all significantly different from zero and greater than 0.30 (ranging from 0.35 to 0.85), confirming the appropriateness of each item as an indicator of the hypothesized latent dimensions.

### 3.2. Multigroup Analysis

#### 3.2.1. Measurement Invariance

First, the hypothesized seven-factor model fit the data reasonably well for both subgroups of employees differing in their working arrangement, that is, traditional workers (N = 326) and remote workers (N = 642) (see Baseline Models in Table 3). As shown in Table 3, multigroup analysis supported measurement invariance up to the scalar level (i.e., factor loadings and intercepts equality holds across groups). Indeed, models M1 (configural model), M2 (metric model) and M3 (scalar model) all adequately fit the data and the differences between the models’ fit indices (M1-M2 and M2-M3 respectively) matched recommended criterions [56].

#### 3.2.2. Structural Invariance and Hypothesis Testing

The posited structural model fit the data adequately for both remote workers (N = 642; χ^2^ (689) = 1250.32, *p* < 0.001; CFI = 0.931; RMSEA [95% CI] = 0.036 [0.032, 0.039]; SRMR = 0.045) and traditional workers (N = 326; χ^2^ (689) = 1100.41, *p* < 0.001; CFI = 0.910; RMSEA [95% CI] 0.042 [0.038–0.047]; SRMR = 0.061). We then tested the unconstrained structural model (i.e., no constraints are imposed on regression paths) and the constrained structural model (i.e., all the coefficient paths are constrained to be equal across groups) for the multi-group comparison between traditional and remote workers. The unconstrained structural model fit the data well (χ^2^ (1407) = 2388.38, *p* < 0.001; CFI = 0.923; RMSEA [95% CI] = 0.038 [0.035, 0.040]; SRMR = 0.053) and served as the baseline against which all subsequently specified models were compared. The constrained model also fit the data (χ^2^ (1451) = 2450.42, *p* < 0.001; CFI = 0.921; RMSEA [95% CI] = 0.038 [0.035, 0.040]; SRMR = 0.057) but it was significantly worse than the unconstrained model (ΔSBχ^2^ (44) = 66,6541, p. 0.015). In other words, some paths were non-invariant across the groups analyzed [55]. Therefore, we performed subsequent iterative tests to the locate non-invariant paths across groups and achieve a final model. The final model, represented in Figure 2, showed an acceptable fit (χ^2^ (1446) = 2422.52, *p* < 0.001; CFI = 0.923; RMSEA [95% CI] = 0.037 [0.035, 0.040]; SRMR = 0.045) and a significantly better fit than the unconstrained baseline model (ΔSBχ^2^ (39) = 75,5844, p. 0.214).

Hypotheses 1 and 2, concerning the role of cross-role interrupting behaviors in directly predicting work–family conflict, were supported in both groups. As expected, NW→W interrupting behaviors were positively related to F→W Conflict (β = 0.359 (0.324)) and unrelated to W→F conflict, while W→NW interrupting behaviors were positively and strongly related to W→F conflict (β = 0.618 (0.593)). In addition, we found W→NW interrupting behaviors to be positively related to F→W conflict too (β = 0.221 (0.194)), albeit much less strongly than the previous relationship. Hypotheses 3a, 3b, 5 and 7, concerning the expected direct relationships between job and personal resources and cross-role interrupting behaviors, were mainly supported in both groups: indeed, NW→W interrupting behaviors were positively associated with job autonomy (β = 0.249 (0.201)) and negatively related to goal-oriented leadership (β = −0.110 (−0.112)), while W→WN interrupting behaviors were positively related to job autonomy and personal initiative at work (β = 0.139 (n.s.) and 0.342 (0.354), respectively). However, we found that the direct path from job autonomy to W→WN interrupting behaviors was non-invariant across groups: contrary to our expectations, job autonomy was unrelated to W→WN interrupting behaviors in the remote worker subgroup.

Hypotheses 4, 6 and 8, regarding the mediating role of cross-role interrupting behaviors between resources and work–family conflict, were partially supported in both groups. Job autonomy was related to F→W conflict both directly and negatively (β = −0.154 (n.s.)) and, as expected, indirectly and positively through NW→W interrupting behaviors (0.042, *p* < 0.001, 95% CI = 0.02, 0.07). Furthermore, we found that the direct path from job autonomy to F→W conflict was non-invariant across groups. As such, NW→W interrupting behaviors fully mediated the relationship between job autonomy and F→W conflict for remote workers, while job autonomy was associated to the outcome through a partially mediated relationship for traditional workers. Moreover, job autonomy was related to W→F conflict both directly and negatively (β = −0.219 (−0.174)) and, as expected, indirectly and positively through NW→W interrupting behaviors (0.073, *p* < 0.05, 95% CI = 0.01, 0.14). However, since the direct path from job autonomy to W→WN interrupting behaviors was non-significant for remote workers, such a partially mediated relationship was only found for traditional workers. As expected, goal-oriented leadership was indirectly and negatively related to F→W conflict through NW→W interrupting behaviors (−0.026, *p* < 0.001, 95% CI = −05., −0.003) and personal initiative at work was indirectly and strongly related to W→F conflict through W→NW interrupting behaviors (0.505, *p* < 0.001, 95% CI = 0.36, 0.67). In other words, cross-role interrupting behaviors fully mediated the relationships between goal-oriented leadership, personal initiative at work and work–family conflict in the expected direction. In addition, we found personal initiative at work to be directly and negatively related with F→W conflict (β = −0.195 (−0.177)), while indirectly and positively related to it through W→NW interrupting behaviors (0.101, *p* < 0.01, 95% CI = 0.05, 0.16), albeit both effects were weaker than the previous and hypothesized paths. As such, personal initiative at work related to F→W conflict through a partially mediated relationship.

We checked the significance of effects with four control variables (i.e., age, gender, education level and professional qualification). In both groups, gender was negatively related to NW→W interrupting behaviors (β = −0.073 (−0.073)). Education level was positively related to job autonomy in the traditional worker subgroup (β = 0.119) and, on the contrary, negatively related in the remote worker subgroup (β = −0.113). In both groups, professional qualification was positively related to job autonomy (β = 0.108 (0.127)), personal initiative at work (β = 0.092 (0.080)) and W→NW interrupting behaviors (β = 0.274 (0.117)). Moreover, professional qualification was positively related to goal-oriented leadership in the traditional worker subgroup (β = 0.160), but not significantly related in the remote worker group. We chose to omit the effects of the control variables from the graphical representation in order to make it clearer.

## 4. Discussion

The integration of the work–family boundary management approach and conservation of resource theory has provided a useful framework for investigating the relationship consequences of work-related resources for work–family conflict. The general aim of this study was to investigate job autonomy, goal-oriented leadership (job resources) and personal initiative at work (work-related personal resource) as antecedents of cross-role interrupting behaviors as well as their relationships with work–family conflict in employees engaged in traditional and remote working arrangements. Our findings substantially confirmed the hypothesized relationships among the study variables. First, our results reveal that both cross-role interrupting behaviors are related to positive increases in work–family conflict experiences.

Furthermore, results show that the relationship between cross-role interrupting behaviors and work–family conflict are mainly directionally congruent: W→NW interrupting behaviors are strongly and positively related to W→F conflict, while NW→W interrupting behaviors are positively related to F→W conflict. As such, the present results corroborate prior research on this relationship [10,31,32,33], which demonstrated that the interrupting role steals time and energy from the interrupted one and, in so doing, the domain that required such role transition is perceived as a generator of interference and strain over the other. Interestingly, we also found that W→NW interrupting behaviors are significantly and positively related to F→W conflict. To the best of our knowledge, we do not know of any studies that have previously found such a result. However, Hunter and colleagues [10] found that boundary violations at home (or work) cause an obstruction of the family (or work) goals attainment, in turn decreasing the personal satisfaction with one’s own resource investment within the family (or work) context. Indeed, the interrupting work (or family) role can cause unscheduled delays in attending to the interrupted family (or work) role, which increases an individual’s rumination over its objectives until it is not resolved [9]. Therefore, it is plausible that workers who often let their private lives be interrupted by work may likely experience a boomerang effect, in which the delay in investment on one’s family role becomes a source of interference concern with one’s job.

Second, the present findings expand the literature regarding the influence exerted by job resources, through the strengthening or loosening of work–family boundaries, on work–family conflict. Results revealed that job autonomy is negatively related to both W→F and F→W conflict and, nevertheless, positively associated with them by increasing W→NW and NW→W interrupting behaviors, respectively. As such, results corroborate prior research on the role of job autonomy in decreasing work–family conflict experiences [16]. However, these results also expand the role blurring hypothesis advanced by previous studies on the contribution of job autonomy to the onset of work–family strain [17,18]. Altogether, these results suggest that autonomy can activate both a resource-enabling and a resource-depleting pathway to work–family conflict. Drawing on COR theory [13], people are motivated to expand or maintain their current resources and yet to achieve this result; they must invest resources. Potentially, if work-life balance is a desirable condition for a worker, then job autonomy may be perceived as a valuable resource to that end. Thanks to job autonomy, he/she owns more freedom to organize and juggle multiple duties by deciding when, where and how to carry out work activities. The flipside of the coin, however, is that such flexibility and such control come at a cost, that is having to allow cross-role interrupting behaviors and their stressful effects. Indeed, when employees benefit from higher autonomy, they are potentially exposed to an intensification of work-life pace and multitasking [18,21], which is often supported by ICT usage [33] and associated with increased pressure to schedule multiple ever-changing demands [65].

Furthermore, findings highlight differences in how such mechanisms operate for remote workers. Results show that, in the remote worker group, (a) NW→W interrupting behaviors fully explains the positive relationship between job autonomy and F→W conflict and (b) job autonomy is not positively related to W→NW interrupting behaviors but is associated with a reduction in W→F conflict. Therefore, these results contribute to the understanding of previous meta-analytic findings by Allen and colleagues [28] that found that flexible work arrangements significantly decrease W→F conflict, while non-significantly affecting F→W conflict. Drawing on Ashforth and colleagues [19], individuals routinize their transitions across roles by incorporating several rites (e.g., emotionally charged symbols like props and clothing) which have the function of signaling and supporting the process to the individual and to role-related members. Since working remotely erases the physical boundaries of the office [20], it may increase the exposure to cues concerning personal or family responsibilities (e.g., home-repair works; duty reminders from relatives), thereby increasing chances for F→W conflict to occur [28]. Contrary to our expectations and to prior literature [6], such a mechanism was not found in the work to non-work direction, so that job autonomy maintains its positive properties. Unlike traditional workers, remote workers’ autonomy is institutionalized and negotiated between the employer and the employee (e.g., on which days is maximized by working remotely and with respect to which objectives), thereby potentially increasing the control over transitions from the family to the work role. In this sense, work-related norms on remote workers’ availability as well as time restrictions occurring during the remote days (e.g., emergencies for which help is required) [47,66] may be way more salient than autonomy itself in triggering W→NW interrupting behaviors.

The present results reveal that goal-oriented leadership is negatively related to F→W conflict by decreasing NW→W interrupting behaviors. This finding expands prior research on the role of supervisor support in decreasing work–family conflict experiences [16,38] by providing evidence for the mediating role of the subordinate’s boundary management. Behaviors such as recognizing the characteristics of the collaborator in assigning goals, delegating, providing feedback and support, enhance motivational resources at one’s disposal to perform successfully and persist in the face of obstacles [22]. Indeed, when employees have positive perceptions of leadership style, they gain additional psychological resources that improve their performance and well-being [13,14]. As such, the employee may benefit from more attentional and self-regulatory resources to resist interrupting work goals to meet family needs, thereby reinforcing boundaries and ameliorating their demanding nature. This result is complementary to a previous study by Ferguson and colleagues [40], which found that supportive supervisor behaviors towards work–family balance increases the subordinate’s ability to stretch work boundaries to attend to family duties which, in turn, facilitate healthier family functioning. Drawing on Border Theory [20], it is plausible that the perceptions of supervisor’s management objectives and style may contribute to the subordinates’ crafting of their boundary management behaviors, thereby differentially shaping work–family outcomes [39].

Regarding personal initiative at work, results revealed that it is positively related to W→F conflict by increasing W→NW interrupting behaviors. This finding corroborates previous results from Bolino & Turnley [25], which found that proactivity at work can increase W→F conflict and highlights the mediating role of cross-role interrupting behaviors in such relationships. Based on prior literature on the resource-depleting side of proactivity at work [24,43], employees with higher personal initiative, by engaging in W→NW interrupting behaviors, may make a resource investment choice–expanding the job role (i.e., gains resources) over the contraction of the family role (i.e., loses resources)–which leaves family boundaries more permeable to work-related interferences and strain. Unexpectedly, we found that personal initiative at work is also negatively associated with F→W conflict and, nevertheless, positively associated to it through W→NW interrupting behaviors. According to Boundary Theory [19], the greater one’s role identification (i.e., the work role for proactive employees), the more likely one is to integrate the role with others (i.e., the family one), thereby forming role transition schemas which ease entries in that role and obstruct exits. It could therefore be hypothesized that proactive employees may devotedly over-drive their resources to work in ways that ultimately aggravate work–family conflict, even though actively solving family demands that interfere with work performance.

### 4.1. Practical Implications

From an applicational standpoint, our results provide several practical implications for promoting healthy and sustainable workplaces. Specifically, we found that job autonomy may be considered a “double-edged sword”, able on the one hand to empower the employee (by making him/her free to organize activities and duties in ways that fit better with his/her needs), but on the other to complicate boundary management between family and work roles. For these reasons, it would be important for organizations to help workers find the right balance in the management of autonomy, supporting them with training courses aimed at enhancing self-awareness about the tactics used to craft one’s autonomy [67]. Along with this, supervisors may be trained on practices that facilitate the mastery of increased autonomy and responsibility, such as goal setting [23].

These implications could be particularly true for employees engaged in new ways of working. In fact, we found that the interruptions arising from non-work domains are so pervasive of remote workers’ experience, such as to completely drain the protective effects of autonomy on the onset of stressful interferences from family at work. Thus, to realize a benefit in terms of the management of work–family roles, it may be important to increase employees’ awareness that boundaries are not solely an individual experience nor static barriers, but that they are constantly negotiated with others. Such awareness may indeed be fundamental for individuals who work remotely in order to learn how to establish clear boundaries and expectations with family and friends [28]. Along with this, we found that goal-oriented leadership style plays a role in the reduction of family-to-work conflict by reducing the occurrence of non-work interruptions at work. Once again this result highlighted that it is fundamental to sensitize leaders to a management style aimed at recognizing, valuing and supporting collaborators to drive their dedication to work and to sustain their well-being. Finally, particular attention is directed to personal initiative at work, which undoubtedly represents a highly sought-after resource for organizations, as it sees employees take on a proactive and pro-company role in the workplace. At the same time, however, this resource seems to have a dark side, namely, difficulty in finding the right balance between being proactive at work without going beyond and interfering with non-work roles.

To prevent personal initiative at work from becoming an obstacle to one’s private life, resulting in increased risks of strain, it may be fundamental to enhance awareness about norms and expectations surrounding proactive employees and to develop a supportive work climate towards personal and family needs [43]. Indeed, individuals have multiple obligations and responsibilities for themselves and to various others in their non-work domain (e.g., spouses, children, relatives and friends), as well as the right to recovery (e.g., leisure time, mastering new skills through hobbies). As such, recognizing those needs in the workplace may constitute the premise for personal fulfilment and motivation of the worker as a whole individual. All in all, organizations and managers should be aware of these potential risks and promote a culture oriented towards the enhancement of human capital, through a family-friendly corporate welfare system [39].

### 4.2. Limitations and Future Research

Some limitations of this research should also be acknowledged. First, the primary limitation derives from the cross-sectional and self-report nature of this research. Experimental, three-waves or even intensive longitudinal designs, which provide temporal separation between variables, should certainly be an objective of future research. Although we have taken some precautions to control for common method bias [54], future research may prospectively integrate other-report data from key domain boundary-keepers (e.g., the spouse in the family domain; supervisors in the work domain) [20]. Second, although this contribution extends the literature regarding new ways of working, it should be noted that we used a dichotomous variable to assess remote working due to sample homogeneity on this variable (i.e., one or two days a week). As such, we did not collect information about the intensity of remote working (i.e., amount of time that employees spend doing tasks away from the conventional work location during the week) [28].

A future research objective should explore how the proposed model differentially applies to employees in organizations characterized by different usage thresholds per week of remote work (e.g., the majority or all their workdays). Third, although this contribution extends the literature on the antecedents and consequences of cross-role interrupting behaviors, future research should address several questions unexplored by the present study. Indeed, we focused on the resource-depleting side of cross-role interrupting behaviors and, nevertheless, scholars have also hinted at possible positive outcomes (e.g., satisfying one’s relatedness need; increased stimulation in routine tasks) [9]. It could be interesting for future research to explore how the present model works when including both positive and negative work–family outcomes [10]. Finally, as this present study highlights the downsides of resources in work–family boundary management, future research should explore the contextual and individual moderating conditions that enable or, conversely, constrain such effects. For example, social norms and expectations regarding availability after work hours [68] and cultural models that valorize intense career commitment and organizational dedication (i.e., work devotion schema) [69] may play a crucial role in offsetting the beneficial effects of resources in work–family boundary management.

On the contrary, higher perceptions of boundary control may help employees to capitalize on job resources, transforming their potential benefits into better work-life outcomes [7].

## 5. Conclusions

This study contributes to the understanding of how personal and job resources are associated with cross-role interrupting behaviors and, in turn, work–family conflict. Specifically, the study highlighted the bright and dark sides of job autonomy, goal-oriented leadership and personal initiative at work in their associations, by positively or negatively relating to the enactment of interrupting behaviors, with work–family strain. Job autonomy and personal initiative at work were found to potentially increase risks of work–family conflict by positively relating to interrupting behaviors. Goal-oriented leadership, on the contrary, was related to decreased non-work interrupting work behaviors, thereby demonstrating an association with decreased family-to-work strain. Moreover, this study found that job autonomy differentially relates to employees’ work–family conflict when engaged in traditional or remote working arrangements. The depleting pathway of job autonomy was not found in the work to non-work direction for remote workers. These results provide meaningful insights on the implications of resources and boundary blurring for employee’s work-life balance, which could be of particular importance for future research and practical purposes.

## Figures and Tables

**Figure 1 ijerph-18-12207-f001:**
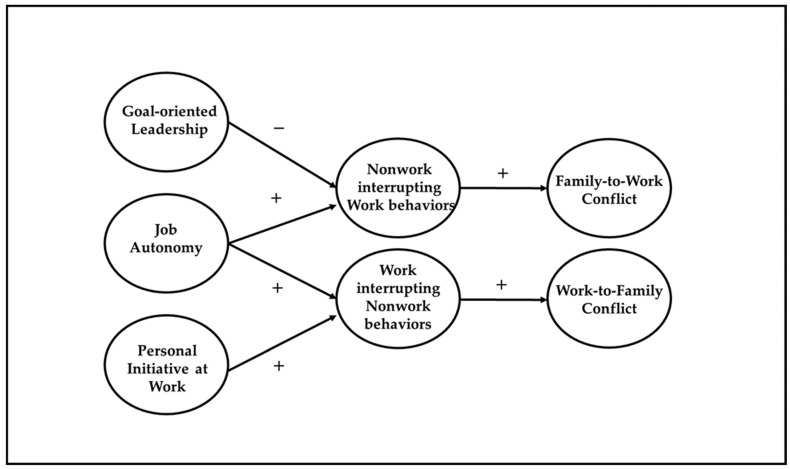
The overarching conceptual model. *Note.* Solid lines are used to represent hypothesized direct effects.

**Figure 2 ijerph-18-12207-f002:**
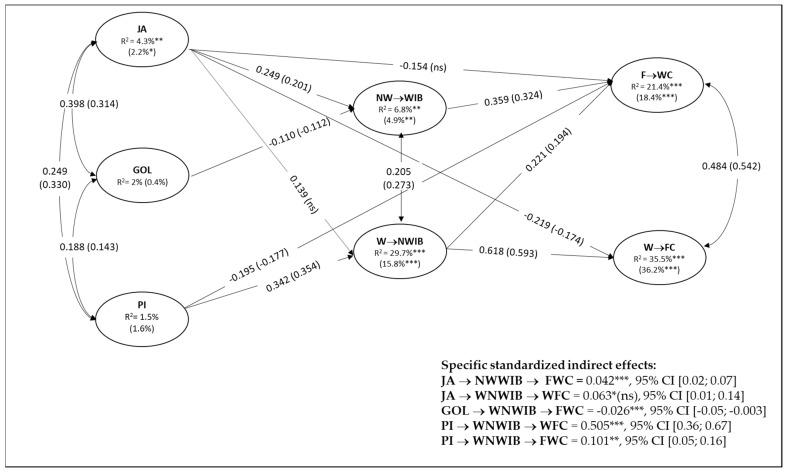
Results from multigroup structural equation analysis and study variables’ explained variance by the model. Notes. * *p* < 0.05; ** *p* < 0.01; *** *p* < 0.001. Figure reports significant standardized regression coefficients between study variables, controlled for gender, age, education level and professional qualification; results for remote workers are reported in parentheses; JA = job autonomy; GOL = goal-oriented leadership; PI = personal initiative at work; NW→WIB = non-work interrupting work behaviors; W→NWIB = work interrupting work behaviors; W→FC = work-to-family conflict; F→WC = family-to-work conflict R^2^ = r-square statistic.

**Table 1 ijerph-18-12207-t001:** Descriptive statistics and zero-order correlations.

		*M*	*SD*	1	2	3	4	5	6	7	8	9	10	11
1.	JA	3.85	1.02	(0.88)										
2.	GOL	3.82	1.07	0.372 **	(0.85)									
3.	PI	4.00	0.47	0.266 **	0.162 **	(0.80)								
4.	NW→WIB	2.51	0.95	0.148 **	−0.034	0.003	(0.72)							
5.	W→NWIB	2.39	1.07	0.193 **	0.142 **	0.331 **	0.218 **	(0.85)						
6.	W→FC	2.14	1.03	−0.044	−0.017	0.094 **	0.146 **	0.510 **	(0.85)					
7.	F→WC	1.57	0.66	0.031	−0.033	−0.097 **	0.349 **	0.178 **	0.436 **	(0.74)				
8.	Gender	1.25	0.44	−0.021	−0.001	0.025	−0.084 *	−0.014	0.009	−0.048	-			
9.	Age	1.72	0.53	−0.113 **	−0.035	0.019	−0.031	−0.011	−0.017	−0.077 *	0.122 **	-		
10.	Edu. Lev.	2.34	0.57	0.031	−0.021	0.088 **	0.033	0.133 **	0.118 **	−0.014	0.185 **	0.029	-	
11.	Prof. Qual.	1.12	0.33	0.114 **	0.060	0.102 **	0.080 *	0.247 **	0.167 **	0.025	−0.020	−0.109 **	0.406 **	-

Notes. * *p* < 0.05; ** *p* < 0.01; coefficient alpha reliability estimates are presented in brackets along the diagonal; M = mean; SD = standard deviation; JA = job autonomy; GOL = goal-oriented leadership; NW→WIB = nonwork interrupting work behaviors; W→NWIB = work interrupting work behaviors; W→FC = work-to-family conflict; F→WC = family-to-work conflict; Edu.Lev. = education level; Prof. Qual. = professional qualification.

**Table 2 ijerph-18-12207-t002:** Results of confirmatory factor analysis and alternative model comparisons.

Models (M)	χ^2^	*df*	CFI	TLI	RMSEA	CI 95%	Δχ^2^
**M0: hypothesized seven-factors model**							
	1538.747	573	0.933	0.926	0.042	0.039, 0.044	
**M1a: 5-factors model** *(W→NWIB and NW→WIB merged; W→FC and F→WC merged)*							
	2746.440	584	0.850	0.838	0.062	0.060, 0.064	1207.693 ** (*df* = 11)
**M1b: 5-factors model** *(W→NWIB and W→FC merged; NW→WIB and F→WC merged)*							
	2942.996	584	0.836	0.823	0.065	0.062, 0.067	1404.249 ** (*df* = 11)
**M2: 1-factor model** *(all 36 items)*							
	10431.048	594	0.317	0.275	0.131	0.129, 0.133	8892.301 ** (*df* = 21)

Notes. ** *p* < 0.001; χ^2^ = chi-square statistic; CFI = comparative fit index; TLI = Tuker–Lewis fit index; RMSEA = root mean square error of approximation; CI = confidence interval; df = degrees of freedom; NW→WIB = nonwork interrupting work behaviors; W→WIB = work interrupting work behaviors; W→FC = work-to-family conflict; F→WC = family-to-work conflict.

**Table 3 ijerph-18-12207-t003:** Results of Tests for Measurement Invariance across Traditional and Remote Workers.

	Model Fit					Model Difference				
Models (M)	χ^2^	*df*	RMSEA (90% CI)	CFI	SRMR	ΔM	Δχ^2^	ΔCFI	ΔRMSEA	ΔSRMR
Baseline Model Traditional Workers	1603.481	573	0.051(0.046, 0.056)	0.907	0.064	−	−	−	−	−
Baseline Model Remote Workers	1237.289	573	0.042 (0.039, 0.046)	0.928	0.048	−	−	−	−	−
M1: Configural	2300.771	1146	0.046 (0.043, 0.048)	0.920	0.054	−	−	−	−	−
M2: Metric	2352.433	1175	0.046 (0.043, 0.048)	0.919	0.056	M1-M2	0.000	−0.001	0.000	0.002
M3: Scalar	2423.320	1204	0.046 (0.043, 0.048)	0.916	0.056	M2-M3	0.000	−0.003	0.000	0.000

Notes. At each step the prior model served as the baseline against which the subsequent specified model was compared in the sequence of invariance tests, all earlier constraints remained in place; χ^2^ = chi-square statistic; RMSEA = Root Mean-Square Error of Approximation; CFI = Comparative Fit Index; SRMR = Standardized Root Mean Square Residual.

## Data Availability

Data sharing not applicable.

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
