# Peer review of "The Bright and Dark Sides of Resources for Cross-Role Interrupting Behaviors and Work–Family Conflict: Preliminary Multigroup Findings on Remote and Traditional Working"

_ijerph, 2021, doi:10.3390/ijerph182212207_

Round 1

Reviewer 1 Report

Overall, this is an interesting study that examines how personal and job resources are associated with cross-role interrupting behaviors and, in turn, work-family conflict. Here are the suggestions for you to consider.

  1. To my understanding, you have considered nonwork interrupting work behavior and work interrupting nonwork behaviors as independent constructs in this study. However, in your statistics, these two constructs are highly correlated. Since they were measured by using different dimensions in the same scale, could it be a measurement issue? It will be helpful if you could provide rationales of why considering them separately to strengthen your arguments and to reduce concerns on conceptualization issues. If it is a measurement issue, please address it in your limitations.
  2. Related to 1., your family-to-work conflict and work-to-family conflict constructs are facing similar challenge. Please consider address this in your arguments and strengthen the connections between your conceptualization, your measurement, and your findings.

Wish you the best!

Reviewer 2 Report

The article concerns the relationship between individual and work resources (goal-oriented leadership, job autonomy and personal initiative at work), nonwork interrupting wok behaviors, work interrupting nonwork behaviors and both family-to-work and work-to-family conflict. The Authors presented a well-structured, theory-based introduction, and a cross sectional study in one sector workers.

To analyse mediation there is a need to conduct either experimental studies or three-wave studies. A cross sectional study design does not allow to draw conclusions on mediation analysis. However, the subject is highly up to date and the results are interesting and important. I propose to change the title: The bright and dark sides of Resources for Cross-role Interrupting Behaviors and Work-Family Conflict: Preliminary findings on remote and traditional working.

I do not feel convinced to conduct zero-ordered correlations for quantitative variables. Dichotomization results in higher correlations and does not represent the variation of variables. Moreover, there is no information on how was the categorisation done (e.g. basing on the median split?). The Authors should explain their reasoning or reanalyse the results.

Concluding, I would like to stress that the article is valuable and the study is worth publishing. Therefore the manuscript needs some minor improvements and then may be published in the International Journal of Environmental Research and Public Health.
